# Viral and cellular telomerase RNAs possess host-specific anti-apoptotic functions

Ahmed Kheimar,[1,2] Laetitia Trapp-Fragnet,[3] Andelé M. Conradie,[1] Luca D. Bertzbach,[1,4] Yu You,[1] Mohammad A. Sabsabi,[1] Benedikt B. Kaufer[1,5]

**ABSTRACT** Human telomerase RNA (hTR) is overexpressed in many cancers and protects T cells from apoptosis in a telomerase-independent manner. The most prevalent cancer in the animal kingdom is caused by the highly oncogenic herpesvirus Marek's disease virus (MDV). MDV encodes a viral telomerase RNA (vTR) that plays a crucial role in MDV-induced tumorigenesis and shares all four conserved functional domains with hTR. In this study, we assessed whether hTR drives tumor formation in this natural model of herpesvirus-induced tumorigenesis. Therefore, we replaced vTR with hTR in the genome of a highly oncogenic MDV. Furthermore, we investigated the anti-apoptotic activity of vTR, hTR, and their counterpart in the chicken [chicken telomerase RNA (cTR)]. hTR was efficiently expressed and did not alter replication of the recombinant virus. Despite its conserved structure, hTR did not complement the loss of vTR in virus-induced tumorigenesis. Strikingly, hTR did not inhibit apoptosis in chicken cells, but efficiently inhibited apoptosis in human cells. Inverse host restriction has been observed for vTR and cTR in human cells. Our data revealed that vTR, cTR, and hTR possess conserved but host-specific anti-apoptotic functions that likely contribute to MDV-induced tumorigenesis.

**IMPORTANCE** hTR is overexpressed in many cancers and used as a cancer biomarker. However, the contribution of hTR to tumorigenesis remains elusive. In this study, we assessed the tumor-promoting properties of hTR using a natural virus/host model of herpesvirus-induced tumorigenesis. This avian herpesvirus encodes a telomerase RNA subunit (vTR) that plays a crucial role in viral tumorigenesis and shares all conserved functional domains with hTR. Our data revealed that vTR and cellular TRs of humans and chickens possess host-specific anti-apoptotic functions. This provides important translational insights into therapeutic strategies, as inhibition of apoptosis is crucial for tumorigenesis.

**KEYWORDS** Marek's disease virus, viral telomerase RNA, human telomerase RNA, apoptosis, tumorigenesis

Telomerase is a ribonucleoprotein complex that is involved in the maintenance of telomeres (1). The enzyme complex consists of two major components, the catalytic subunit telomerase reverse transcriptase (TERT) and the telomerase RNA (TR). TRs are constitutively expressed and have a highly conserved secondary structure in vertebrates including humans and chickens (2, 3). Intriguingly, human telomerase RNA (hTR) is overexpressed in many cancers (4), including lung cancer (5), prostate cancer (6), and leukemia (7). Furthermore, it has recently been demonstrated that hTR inhibits apoptosis in human T cells independently of its role in the telomerase complex (8). Similarly, a knockdown of hTR in cancer cells using siRNAs induces apoptosis (9).

Address correspondence to Ahmed Kheimar, ahmed1985@zedat.fu-berlin.de, or Benedikt B. Kaufer, b.kaufer@fu-berlin.de.

The authors declare no conflict of interest.

See the funding table on p. 11.

Marek's disease virus (MDV) is a highly oncogenic alphaherpesvirus that causes T-cell lymphoma in chickens and represents a natural small animal model for herpesvirus-induced tumorigenesis (10). MDV encodes a viral TR homologue [viral telomerase RNA (vTR)] that plays a crucial role in MDV-induced tumorigenesis (11); however, the underlying mechanism remains poorly understood. vTR is the most abundant viral transcript in MDV-induced tumor cells (11, 12) and shares the conserved functional domains with cellular TRs (13). We recently demonstrated that overexpression of chicken TR (cTR) complements the loss of vTR in MDV-induced tumorigenesis (14), highlighting that high cTR expression levels can drive tumorigenesis. The secondary structures of vTR, cTR, and hTR are highly conserved, including the pseudoknot (core) domain that harbors the template sequence (CR-1) for the extension of telomeric repeats (15), the CR4-CR5 domain that is essential for proper assembly with TERT (15) and the H/ACA box and CR-7 domain, which are responsible for TR stability and localization (13, 16, 17). In addition, we showed that both vTR and hTR interact with chicken TERT and mediate telomerase activity *in vitro* (18). Therefore, we investigated whether overexpression of hTR could complement the loss of vTR in MDV-induced tumorigenesis and whether vTR possesses anti-apoptotic functions.

In this study, we generated a recombinant MDV expressing hTR instead of vTR. This recombinant virus efficiently expressed hTR in infected cells and replicated in a manner comparable to that of the wild-type (wt) virus. Interestingly, hTR did not compensate for the loss of vTR in MVD-induced tumorigenesis. Because inhibition of apoptosis likely contributes to tumor formation, we assessed the ability of hTR, vTR, and cTR to inhibit apoptosis in human and chicken cells. Strikingly, hTR did not inhibit apoptosis in chicken cells despite its high activity in human cells. We also provide the first evidence that vTR and cTR possess anti-apoptotic functions, but only in chicken cells and not in human cells. Taken together, our study revealed that viral, chicken, and human TRs share common anti-apoptotic functions that are not conserved across the two species.

## RESULTS

### Generation and characterization of recombinant MDV expressing hTR

To determine whether hTR possesses tumor-promoting functions and could complement the loss of vTR in MDV-induced tumorigenesis, we generated a recombinant virus expressing hTR (vhTR) instead of vTR (Fig. 1A). vTR was first deleted in the MDV genome (vΔvTR), and then hTR was inserted downstream of the native vTR promoter. The resulting clones were screened by restriction fragment length polymorphism (RFLP), PCR, and Sanger sequencing, and the entire viral genome was confirmed by Illumina MiSeq sequencing (1,000-fold coverage). To determine whether deletion of the vTR or insertion of the hTR affects viral replication, we assessed the replication properties of the recombinant viruses. Plaque size assays revealed that vhTR replicated in a manner comparable to that of the wt virus (Fig. 1B). The replication properties were confirmed by multi-step growth kinetics (Fig. 1C), highlighting that neither deletion of vTR nor insertion of hTR alters MDV replication *in vitro*.

### hTR is efficiently expressed in vhTR-infected cells

To confirm the expression of the respective TRs, we infected chicken embryo cells (CECs) with vhTR, vΔvTR, or wt virus and performed RT-qPCR. As anticipated, vTR was only expressed in cells infected with the wt virus, whereas its expression was completely abrogated in vΔvTR and vhTR (Fig. 2A). Consistently, hTR was efficiently expressed in vhTR-infected cells at levels comparable to vTR in the wt virus (Fig. 2B).

### Overexpression of hTR does not complement the loss of vTR *in vivo*

To determine whether hTR possesses tumor-promoting properties, we infected 1-day-old chickens with 2,000 plaque-forming units (PFU) of either wt, vΔvTR, or vhTR. Quantitative PCR (qPCR) analyses revealed that vΔvTR and vhTR replicated comparably to the wt virus

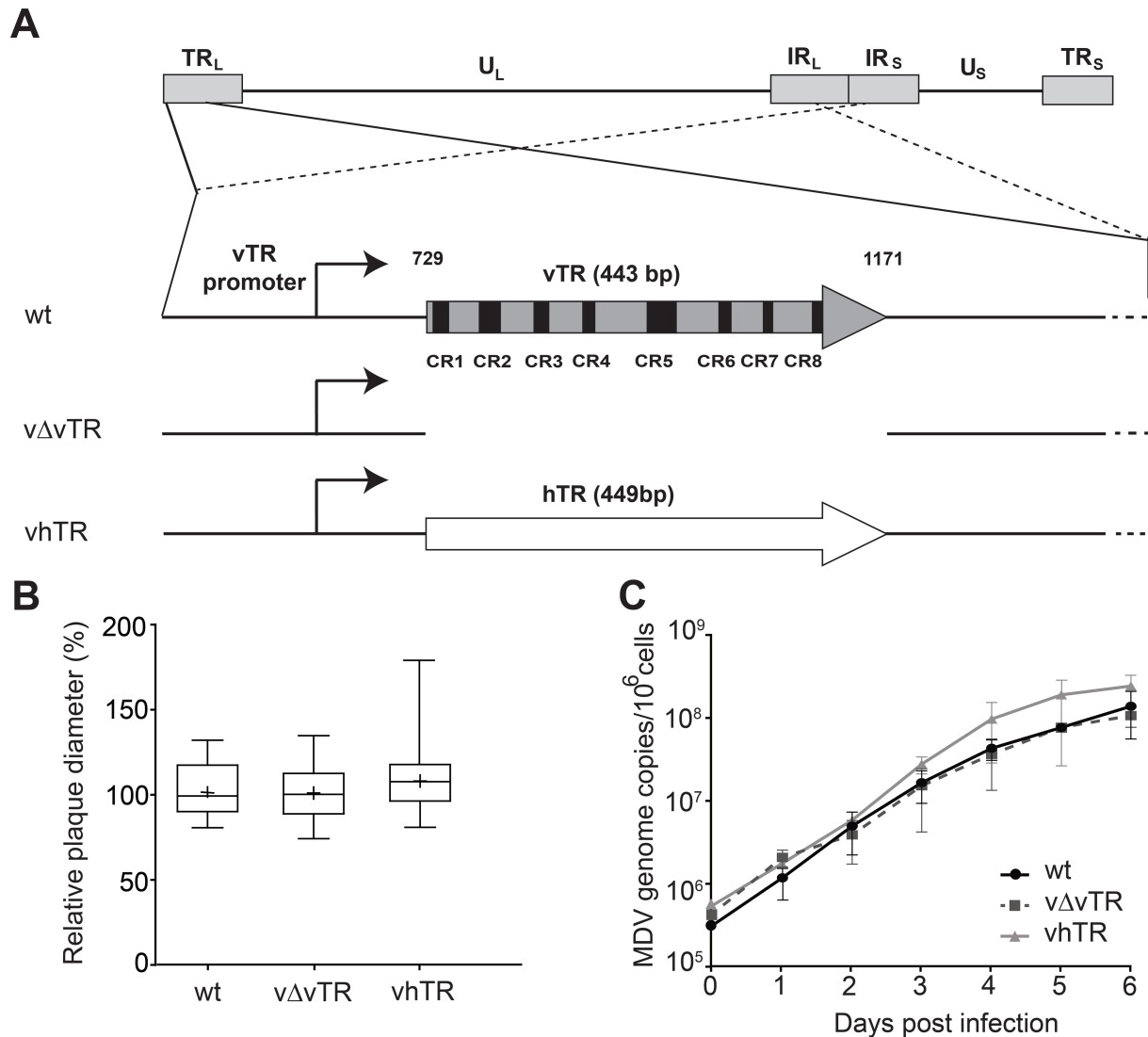

FIG 1 Generation and characterization of the recombinant MDV mutants. (A) Overview of the MDV genome with a focus on the vTR with its conserved regions (CR1– CR8). Recombinant viruses that either lack the entire vTR (vΔvTR) or encode human TR (vhTR) instead of vTR are shown below. (B) Plaque size assays of indicated recombinant viruses. The plaque sizes are shown as box plots with minimums and maximums. Results are shown as the mean of three independent experiments. No significant differences were detected compared to wt ($P > 0.05$; one-way analysis of variance, $n = 150$). (C) Multi-step growth kinetics of indicated viruses. Mean MDV genome copies per million cells are shown for the indicated time points. The chicken inducible nitric oxide synthase (iNOS) gene was used for normalization. Mean genome copies of one independent experiment in triplicates are shown with standard deviations (error bars). No significant differences were detected compared to wt ($P > 0.05$, Kruskal-Wallis test).

in the blood of infected animals (Fig. 3A). Over the course of infection, the animals were monitored daily for clinical symptoms. Intriguingly, disease incidence was reduced in animals infected with vΔvTR and vhTR when compared to those infected with the wt virus (Fig. 3B). Furthermore, tumor incidence was significantly reduced in both vΔvTR- and vhTR-infected animals when compared to the wt virus (Fig. 3C), indicating that hTR neither complements vTR nor has tumor-promoting properties in this natural model of virus-induced tumor formation. To assess the potential effects of hTR on tumor dissemi- nation, visceral organs with gross tumors were quantified during necropsy. While wt virus-induced tumors disseminated to various organs, vhTR and vΔvTR tumors were restricted to a single organ per animal (Fig. 3D). To ensure that hTR was efficiently expressed in vhTR-induced tumors, we performed RT-qPCR on the tumor tissues. RT- qPCR revealed that hTR was efficiently expressed in vhTR-induced tumors at levels

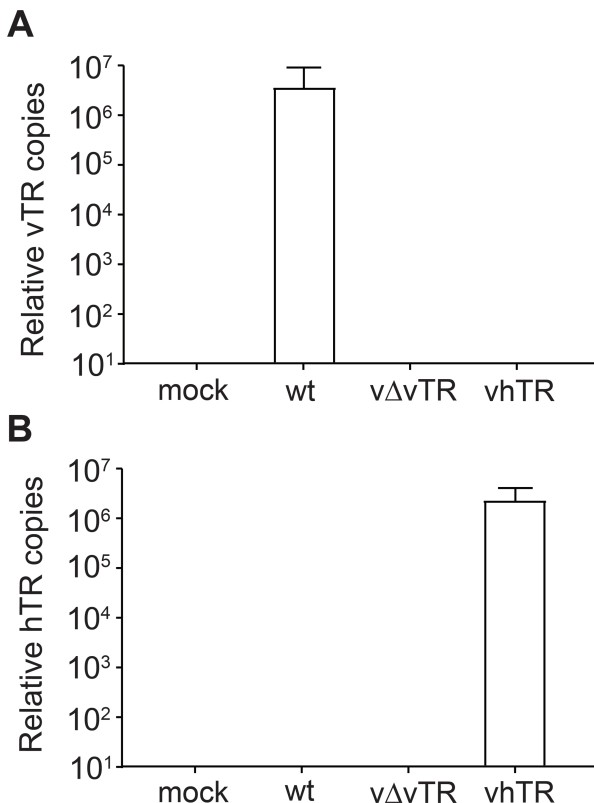

**FIG 2** Quantification of vTR and hTR expressions in MDV-infected cells. One million CECs were infected with 1,000 PFU of indicated viruses; total RNAs were isolated at 6 dpi and RT-qPCR was performed. The mean copy numbers of (A) vTR and (B) hTR are shown for corresponding viruses relative to the expression levels of the cellular glyceraldehyde 3-phosphate dehydrogenase (GAPDH) ($P > 0.05$, Kruskal-Wallis test). Results are shown as means of three independent experiments with standard errors (error bars).

comparable to vTR in wt virus tumors (Fig. 4A and B). Our data demonstrate that hTR does not drive tumor formation in our natural virus-host model of herpesvirus-induced tumorigenesis.

## TRs possess common anti-apoptotic functions that are host specific

To determine why hTR does not complement the loss of vTR in infected chickens, we assessed the anti-apoptotic activities of hTR, vTR, and cTR. To date, only hTR has been shown to possess anti-apoptotic activity in studies using human cells; however, it remains unclear whether this activity is conserved across species between chickens and humans. In addition, it remains unknown whether vTR can inhibit apoptosis, which could contribute to MDV-induced tumorigenesis. To investigate the TR-mediated anti-apoptotic activity in chicken T cells, we generated CU91 T cells expressing hTR, cTR, vTR, or an empty vector. Apoptosis was induced using 1-µM staurosporine (STS), and apoptosis levels were measured using Caspase-Glo 3/Caspase-Glo 7, 8, and 9 assays. Strikingly, vTR and cTR significantly inhibited apoptosis in chicken T cells, whereas no inhibition was detected for hTR (Fig. 5A through C). To confirm these findings, we used another chicken T-cell line, called 855–19. As observed in CU91 cells, vTR and cTR significantly inhibited STS-induced apoptosis, whereas hTR did not (Fig. 5D through F). In addition, we assessed the anti-apoptotic activity of the TRs in human 293T cells expressing hTR, cTR, or vTR. As expected, hTR efficiently inhibited apoptosis in STS-treated 293T cells, whereas no inhibition was observed in 293T cells expressing vTR or cTR (Fig. 5G through I). To ensure that the observed effects were not due to differences in the expression levels, we performed RT-qPCR. vTR, cTR, and hTR were efficiently expressed in

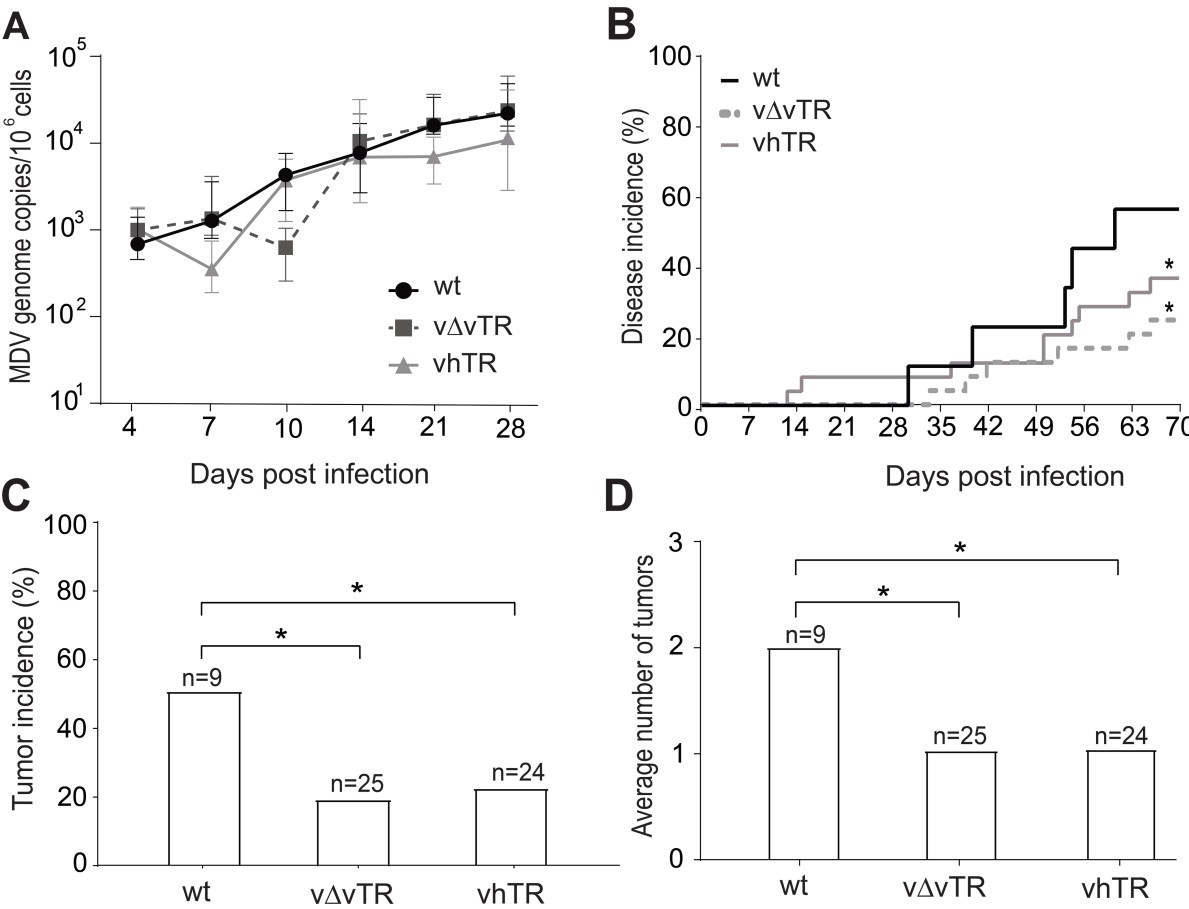

**FIG 3** hTR does not complement the loss of vTR in MDV-induced tumorigenesis. (A) qPCR detecting MDV genome copies in the blood of chickens infected with wt (*n* = 9), vΔvTR (*n* = 25), and vhTR (*n* = 24). Mean MDV genome copies per one million cells are shown for the indicated time points. No significant differences were detected compared to wt (*P* > 0.05, Kruskal-Wallis test). (B) Disease incidence is shown as a percentage of animals that developed clinical symptoms along the course of infection. Significant differences in comparison to wt are indicated by an asterisk (*P* < 0.0125, Fisher's exact test). (C) Tumor incidence in chickens infected with indicated viruses is shown as a percentage of animals developing gross tumors. Significant differences are indicated by an asterisk (*P* < 0.0125, Fisher's exact test). (D) Mean number of visceral organs with macroscopic tumors per animal infected with the indicated viruses. Significant differences are indicated by an asterisk (*P* < 0.0125, Fisher's exact test).

the respective CU91, 855–19 T, and 293T cells (Fig. 6A through C). Taken together, our data revealed that viral and cellular TRs share common anti-apoptotic properties that are not conserved between humans and chickens. The observed anti-apoptotic activity of vTR likely plays an important role in the tumorigenesis of this highly oncogenic herpesvirus.

## DISCUSSION

To date, the role of TRs in tumorigenesis has mainly been restricted to their role in the telomerase complex. hTR has recently been shown to possess anti-apoptotic activity independent of its role in the telomerase complex (8). However, it remains unclear whether the anti-apoptotic activity of hTR drives oncogenesis. Viruses have evolved several strategies to block apoptosis and prolong the survival of infected cells (19). The highly oncogenic MDV encodes a TR homologue that plays an important role in virus-induced tumorigenesis; however, the underlying mechanism remains poorly understood (11). We recently demonstrated that a vTR mutant (vTR P6.1) that is not incorporated into the telomerase complex efficiently promotes tumor formation (20). This suggests that vTR possesses pro-oncogenic function(s) that are independent of its telomerase activity. Since TRs share the main functional domains, we investigated whether hTR also

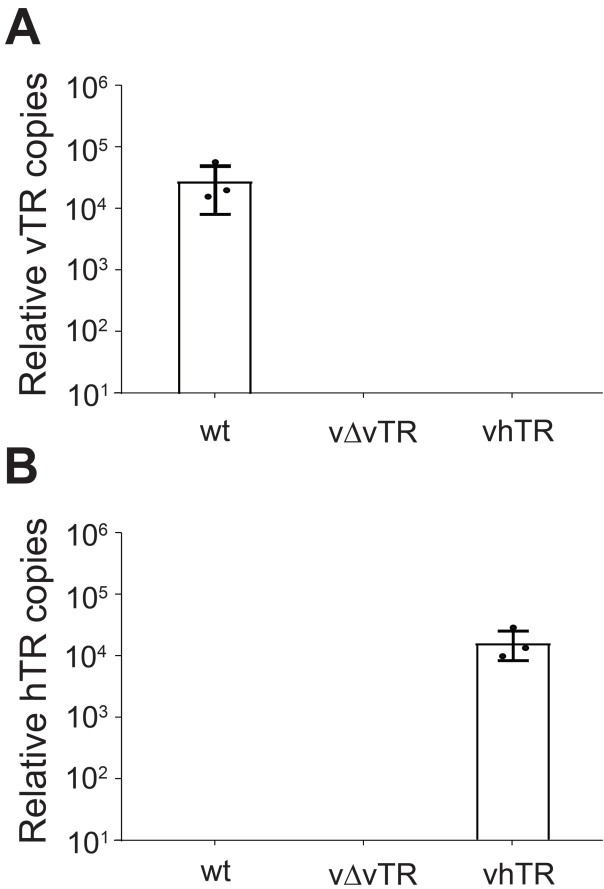

**FIG 4** Quantification of vTR and hTR expression levels in MDV-induced tumors. RT-qPCR analysis of (A) vTR and (B) hTR expressions in tumor tissue derived from chickens infected with indicated viruses. Results are shown as the mean of three tumor tissues for each group normalized to cellular GAPDH with standard errors (error bars) ($P > 0.05$, Kruskal-Wallis test).

possesses pro-oncogenic properties in our natural virus-host model of virus-induced tumorigenesis. We generated a recombinant virus expressing hTR instead of vTR. Neither vTR deletion nor hTR insertion altered viral replication *in vitro* or *in vivo*. This supports our previous finding that vTR is dispensable for MDV replication (14, 21). To confirm the efficient expression of hTR in vhTR-infected cells, we used RT-qPCR. Our data revealed that hTR was efficiently expressed in the vhTR-infected cells. This expression was comparable to that of vTR in the wt virus, as both vTR and hTR were driven by the same promoter in the viral genome. To investigate whether hTR could complement the loss of vTR in MDV-induced oncogenesis, we infected chickens with the recombinant viruses. Disease development was severely impaired in the absence of vTR, as described previously for partial and complete deletions of vTR (11, 14). Intriguingly, hTR was not able to restore the oncogenic activity of vΔvTR, as both disease and tumor incidence were severely impaired compared to the wt virus. This was surprising, as the insertion of the cellular TR from the chicken (cTR) completely restored the tumorigenesis of the virus (14). This demonstrated that hTR does not drive tumorigenesis in this natural model of virus-induced tumorigenesis, even though hTR was efficiently expressed in the respective tumors.

Due to the Covid-19 pandemic, we had to terminate the hTR animal experiment earlier (70 dpi) compared to the previous cTR animal experiment (91 days). To address the differences in the timescale, we compared the tumor incidence induced by vcTR and vhTR at 70 dpi. Our analyses revealed that cTR significantly increased tumor incidence compared to vΔvTR. This was not observed for vhTR, suggesting that hTR does not

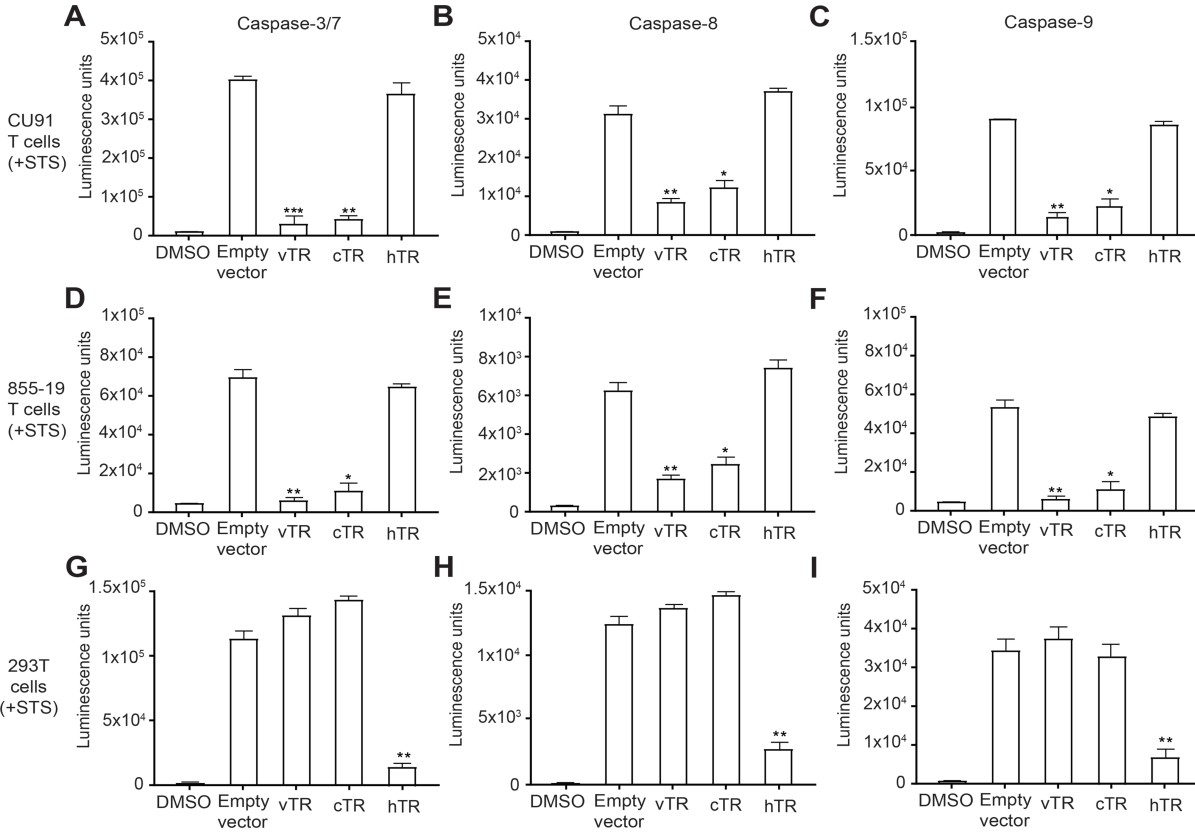

**FIG 5** hTR, cTR, and vTR exhibit anti-apoptotic activity in their respective host cells. The activation of the caspase pathway in 293T cells and chicken T cells (CU91 and 855–19 T) using 1-μM STS. The caspase activity was measured in different cell lines, and caspase 3/7, caspase 8, and caspase 9 are shown as luminescence units for (A–C) chicken T cells CU91, (D–F) chicken T cells 855–19 T, and (G–I) 293T cells. All CU91, 855–19 T, and 293T cells harboring vTR, cTR, or hTR sequences were cultured with 1-μM STS (except the DMSO control) for 3 h at 40°C or 37°C. Luminescence was normalized to background levels. Error bars represent the standard error of the mean of three independent experiments performed in duplicates. Asterisks indicate significant differences in comparison to the empty vector control (*$P < 0.05$, **$P < 0.01$, and ***$P < 0.001$ assessed by one-way analysis of variance).

complement the loss of vTR in MDV-induced tumorigenesis and that the differences in the timescale have no impact on the tumor incidence induced by vhTR. One potential explanation for this could be that hTR is unable to reconstitute a functional telomerase complex with the chicken TERT. This seems less likely, as we previously demonstrated that vTR and hTR efficiently reconstituted telomerase activity with chicken TERT *in vitro* (18). In addition to TR and TERT, many telomerase-associated proteins are required for the assembly of a functional complex at telomeres in living cells, which in turn could contribute to the observed phenotype. A second explanation for the inability of hTR to drive tumorigenesis in chickens may be the improper expression of hTR in vhTR-induced tumors. However, RT-qPCR revealed that hTR was efficiently expressed in the vhTR-induced tumors.

The third explanation is the anti-apoptotic properties of hTR. Therefore, we investigated whether hTR, cTR, or vTR could protect chicken cells from apoptosis. We generated polyclonal CU91 and 855–19 chicken T-cell lines expressing hTR, cTR, or vTR, confirmed their TR expression levels, and assessed their anti-apoptotic activity. Here, we demonstrated for the first time that vTR and cTR possess anti-apoptotic activities in chicken T cells, which are the primary target cells of MDV infection and tumorigenesis. Future studies will further investigate the vTR-mediated anti-apoptotic mechanism(s) and determine whether the activity is dependent or independent of its role in the telomere complex.

In contrast, hTR did not inhibit apoptosis in chicken T cells. This could explain why hTR did not drive tumorigenesis in our model as it lacks the ability to inhibit apoptosis in chicken T cells. To assess whether vTR or cTR could also inhibit apoptosis in human cells,

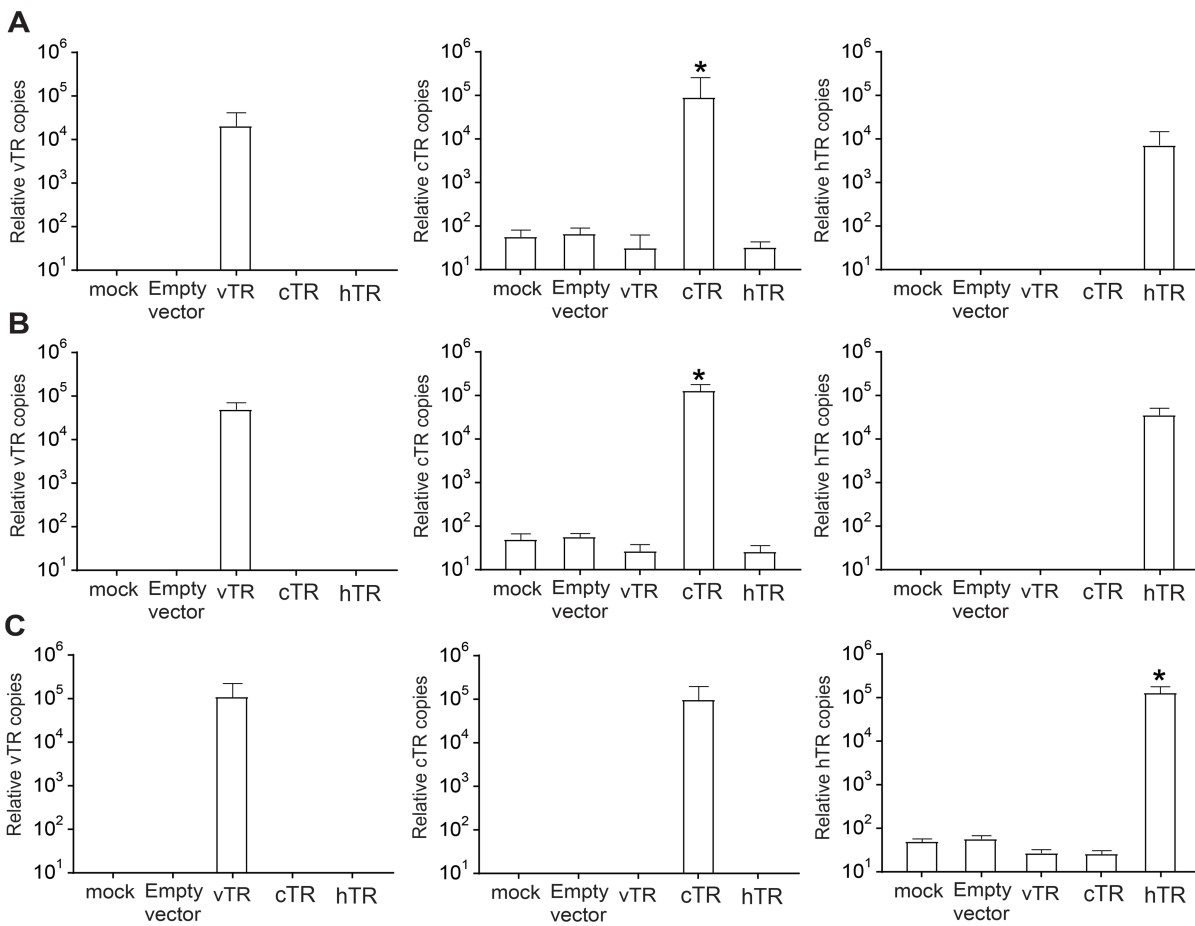

**FIG 6** Quantification of TRs expression levels in CU91, 855–19 T cells, and 293T. Total RNAs were extracted from polyclonal cells expressing vTR, cTR, or hTR and their expression quantified by RT-qPCR. (A) vTR, cTR, and hTR expression levels in CU91 relative to the expression levels of cellular GAPDH ($P > 0.05$, Kruskal-Wallis test). (B) vTR, cTR, and hTR expression levels in 855–19 T relative to the expression levels of cellular GAPDH ($P > 0.05$, Kruskal-Wallis test). (C) vTR, cTR, and hTR expressions in 293T cells are shown relative to the expression levels of cellular β2 microglobulin. Asterisks indicate the significant differences in comparison to the basal expression of cTR or hTR of the respective cells (*$P < 0.05$, Kruskal-Wallis test).

we induced apoptosis in 293T cells expressing hTR, cTR, or vTR. Caspase activity assays revealed that hTR protected 293T cells from apoptosis, whereas vTR and cTR did not. This revealed that TRs share anti-apoptotic functions; however, the interaction partners and/or the mechanism(s) are not conserved enough to function across these two distantly related species.

## MATERIALS AND METHODS

### Cells

CECs were prepared from Valo specific-pathogen-free (SPF) embryos (Valo BioMedia) and maintained as described previously (22). Reticuloendotheliosis virus-transformed chicken T-cell lines CU91 and 855–19 T cells (23, 24) were grown in suspension in RPMI 1640 medium supplemented with 1% sodium pyruvate, 1% non-essential amino acids, 10% fetal bovine serum, and 1% penicillin (100 U/mL)/streptomycin (100 µg/mL), and maintained at 41°C in a 5% $CO_2$ atmosphere.

### Generation of hTR expressing viruses

Recombinant MDV encoding hTR instead of vTR was generated using a bacterial artificial chromosome (BAC) of the very virulent RB-1B strain (pRB-1BΔIRL) by en passant

mutagenesis, as described previously (22, 25). Briefly, the hTR gene (AF221907) (13) was inserted into a previously generated recombinant MDV with a complete deletion of the vTR sequence (vΔvTR), resulting in the vhTR virus (14, 21). The resulting BAC clones were validated by RFLP using multiple restriction enzymes, PCR, and Illumina MiSeq sequencing. The primers used for mutagenesis and sequencing are listed in Table 1. The recombinant viruses were reconstituted by calcium transfection of CECs with recombinant BAC DNA as described previously (25, 26).

## Quantification of vTR and hTR expression *in vitro* and *in vivo*

vTR and hTR expression levels were quantified in MDV-infected cells and tumor tissues using RT-qPCR. Briefly, $10^6$ CECs were infected with $10^3$ PFU of wt, vΔvTR, or vhTR. Total RNA was isolated from infected CECs and tumor tissues using the RNeasy Plus Mini Kit (Qiagen) and the innuPREP Virus RNA kit (Analytik Jena GmbH), respectively, according to the manufacturer's instructions. The isolated RNAs were treated with RQ1 RNase-Free DNase (Promega), and reverse transcription was performed using the High-Capacity cDNA Reverse Transcription Kit (Applied Biosystems). vTR and hTR expression was quantified using TaqMan qPCR, and the cellular GAPDH reference gene was used to normalize vTR and hTR expression. The vTR, hTR, and GADPH primer and probe sets have been previously published (14, 21) and are shown in Table 1.

**TABLE 1** Primers and probes for RT-qPCR, qPCR, DNA sequencing, and construction of the recombinant viruses

| Construct | | Sequence (5′ → 3′) |
|---|---|---|
| vΔvTR | For | CGGAGGAAGCTACAAGAGCCCCACGCGGGGTTCCCCCGGCGCGGCCCCGCGCGCACGACCT AGGGATAACAGG-GTAATCGATTT |
| | Rev | TCTACTCACAGAGCCCCGCGCGCGGCTCAACGGCTCCAACGGTCGTGCGCGCGGGGCCGCGCCAGTGTTACAACC-AATTAACC |
| vhTR | For | CGGAGGAAGCTACAAGAGCCCCACGCGGGGTTCCCCCGGCGTTGCGGAGGGTGGGCCTGG |
| | Rev | CGCGGCTCAACGGCTCCAACGGTCGTGCGCGCGGGGCCGCGCATGTGTGAGCCGAGTCCTGGG |
| vTR locus | For | GCCCCTCTCTGCTCGCTCT |
| | Rev | CACACGTCCAGGCCAGGA |
| pvTR-TRs into pLKO5 vector | For | ACTAT<u>CATATG</u>[a]GACAGGCAGTTGTACACCTGCCTG |
| | Rev | ATCTA<u>GAATTC</u>[a]TGCGCGCGGGGCCGC |
| vTR (RT-qPCR) | For | CCTAATCGGAGGTATT GATGGTACTG |
| | Rev | CCCTAGCCCGCTGAAAGTC |
| | Probe | FAM-CCCTCCGCCCGCTGTTTACTCG-TAM |
| hTR (RT-qPCR) | For | GGTGGTGGCCATTTTTTGTC |
| | Rev | CTAGAATGAACGGTGGAAGGC |
| | Probe | FAM-CGCGCTGTTTTTCTCGCTGACTTTC-TAM |
| cTR (RT-qPCR) | For | TGGAAGGCTCCGCTGTGC |
| | Rev | GGAGCGCGGCGACAGC |
| | Probe | FAM-CTAATCGGGGGGAATTGATGG -TAM |
| ICP4 (qPCR) | For | CGTGTTTTCCGGCATGTG |
| | Rev | TCCCATACCAATCCTCATCCA |
| | Probe | FAM-CCCCCACCAGGTGCAGGCA-TAM |
| iNOS (qPCR) | For | GAGTGGTTTAAGGAGTTGGATCTGA |
| | Rev | TTCCAGACCTCCCACCTCAA |
| | Probe | FAM-CTCTGCCTGCTGTTGCCAACATGC-TAM |
| GAPDH (RT-qPCR) | For | GAAGCTTACTGGAATGGCTTTCC |
| | Rev | GGCAGGTCAGGTGAACAACA |
| | Probe | FAM-TGTGCCAACCCCCAAT-TAM |
| ß2M (RT-qPCR) | For | CCAGCAGAGAATGGAAAGTCAA |
| | Rev | TCTCCATTCTTCAGTAAGTCAACTTCA |
| | Probe | FAM-ATGTGTCTGGGTTTCATCCATCCGACA-TAM |

[a]Restriction enzyme sites are underlined.

## Plaque size assays and growth kinetics

The replication properties of the recombinant viruses were assessed using plaque size assays and multi-step growth kinetics as described previously (27). Briefly, $10^6$ CECs were infected with 100 PFU of either wt, vΔvTR, or vhTR virus. Infected cells were fixed at 6 days post-infection (dpi), and plaque areas were measured using the Bioreader System (Bio-Sys; Karben, Germany), from which plaque diameters were calculated. For multi-step growth kinetics, infected cells were harvested daily for 6 days; viral DNA was extracted using the RTP DNA/RNA virus Mini kit (Stratec; Berlin, Germany), and viral copies were quantified by qPCR. The viral genome copies were normalized to the chicken inducible nitric oxide synthase (iNOS) gene. Primers and probes specific for the MDV immediate-early gene ICP4 and the cellular iNOS gene are shown in Table 1.

## Animal experiment

One-day-old male and female SPF chickens (Valo BioMedia) were randomly assigned to three groups. Animals were infected subcutaneously with 2,000 PFU of either wt ($n = 9$), vΔvTR ($n = 25$), or vhTR ($n = 24$). Peripheral blood was collected from infected chickens at 4, 7, 10, 14, 21, and 28 dpi to determine MDV genome copies in the blood, as previously described (26, 28). Chickens were monitored daily for clinical symptoms throughout the experiment. The animal experiment was performed in a blinded manner to eliminate bias and avoid subjectivity in the assessment of clinical symptoms. The animals were euthanized and examined for tumor lesions once the clinical symptoms were evident or at 70 dpi after the termination of the experiment. To confirm the presence of the introduced mutations in the viral genome, DNA was extracted from the tumor tissues, and the target region was analyzed by Sanger sequencing.

## Quantification of virus genome copies

DNA was isolated from the blood samples of the infected animals using the DNA E-Z96 blood kit (Omega Biotek, USA). Virus genome copies were assessed by qPCR using primers and probes specific for ICP4 and iNOS (Table 1) as described above (26, 28). Viral genome copies were normalized to the iNOS gene, as previously described (29).

## Lentivirus production and generation of clonal cell lines

To facilitate the expression of vTR, cTR, or hTR in chicken T cells, we generated lentiviral vectors using the pLKO5.sgRNA.EFS.PAC vector (a gift from Benjamin Ebert; Addgene, #57825) (30). The vTR, cTR, or hTR sequences, including the native vTR promoter, were amplified by PCR from the corresponding recombinant viruses and inserted into the pLKO5 vector between the NdeI and EcoRI sites. The primers used for cloning and sequencing are listed in Table 1. TR-expressing lentiviruses were generated by transfecting 293T cells with pCMVDR8.91, pCMV-VSV-G, pLKO5.vTR, pLKO5.cTR, pLKO5.hTR, or the empty vector control. Lentiviruses were harvested 36 h post-transfection and used for transduction as described previously (31). After transduction of CU91, 855–19 T cells, and 293T cells, cells were selected with 1- or 2-µg/mL puromycin, respectively, for 5 days. The expression levels of vTR, cTR, and hTR in the polyclonal cells were confirmed by RT-qPCR.

## Apoptosis assays

Apoptosis was induced in 293T or chicken T cells using 1-µM STS (Sigma-Aldrich), and dimethyl sulfoxide (DMSO) was used as a control. The level of apoptosis was evaluated using the Caspase-Glo 3/Caspase-Glo 7, Caspase-Glo 8, and Caspase-Glo 9 assay systems (Promega) according to the manufacturer's instructions. Briefly, cells were seeded at a density of $5 \times 10^4$ cells/well in 96-well plates and incubated with STS for 3 h, after which 100-µL Caspase-Glo reagent was added to the cells. After mixing, cells were transferred to a 96-well white-walled plate and incubated at room temperature in the dark for 30 min. Luminescence of the samples was measured using a Synergy H1 Hybrid Reader

(BioTek, Germany). Relative light units were calculated by subtracting the blank control (media only) values.

## Statistical analyses

Statistical analyses were performed using the SPSS software (SPSS, Inc.) and GraphPad Prism v.7. The data were first analyzed for normal distribution. Plaque size assays were performed using one-way analysis of variance. qPCR data were evaluated using the Kruskal-Wallis test. The results were considered statistically significant at $P < 0.05$. Fisher's exact test and Kaplan-Meier survival analysis using log-rank test (Mantel-Cox test) were used for animal experimental data as indicated, with the Bonferroni correction for multiple comparisons, and were considered significant if $P < 0.0125$.

## ACKNOWLEDGMENTS

We are grateful to Ann Reum (Institute of Virology, Freie Universität Berlin) and Marion Müller and Yvonne Weber (Institute of Immunology, Freie Universität Berlin) for their technical assistance. The authors thank all lab members for the helpful discussions.

This work was supported by the Deutsche Forschungsgemeinschaft grant KA 3492/8–1 awarded to B.B.K.

All authors critically read and revised the manuscript and approved the final version.

The authors declare that they have no known competing financial interests or personal relationships that could have appeared to influence the work reported in this paper.

## AUTHOR AFFILIATIONS

[1]Institute of Virology, Freie Universität Berlin, Berlin, Germany

[2]Department of Poultry Diseases, Faculty of Veterinary Medicine, Sohag University, Sohag, Egypt

[3]INRAE, UMR1282 Infectiologie et Santé Publique, Equipe Biologie des Virus Aviaires INRAE, Nouzilly, France

[4]Department of Viral Transformation, Leibniz Institute of Virology (LIV), Hamburg, Germany

[5]Veterinary Centre for Resistance Research (TZR), Freie Universität Berlin, Berlin, Germany

## AUTHOR ORCIDs

Ahmed Kheimar  http://orcid.org/0000-0001-8121-8981
Andelé M. Conradie  http://orcid.org/0000-0001-9975-0642
Luca D. Bertzbach  http://orcid.org/0000-0002-0698-5395
Benedikt B. Kaufer  http://orcid.org/0000-0003-1328-2695

## FUNDING

| Funder | Grant(s) | Author(s) |
| --- | --- | --- |
| Deutsche Forschungsgemeinschaft (DFG) | KA 3492/8-1 | Benedikt B. Kaufer |

## AUTHOR CONTRIBUTIONS

Ahmed Kheimar, Conceptualization, Data curation, Investigation, Methodology, Writing – original draft | Laetitia Trapp-Fragnet, Investigation, Resources, Writing – review and editing | Andelé M. Conradie, Investigation, Resources, Writing – review and editing | Luca D. Bertzbach, Conceptualization, Resources, Writing – review and editing | Yu You, Conceptualization, Investigation, Writing – review and editing | Mohammad A. Sabsabi, Conceptualization, Investigation, Writing – review and editing | Benedikt B. Kaufer,

Conceptualization, Funding acquisition, Project administration, Resources, Supervision, Writing – original draft, Writing – review and editing

## DATA AVAILABILITY

All relevant data are presented in this study and are available from the corresponding authors upon reasonable request.

## ETHICS APPROVAL

All animal experiments were conducted in accordance with German and European guidelines and were approved by the governmental authority, the Landesamt für Gesundheit und Soziales of Berlin, Germany (approval number G0218/12).

## ADDITIONAL FILES

The following material is available online.

### Open Peer Review

**PEER REVIEW HISTORY (review-history.pdf).** An accounting of the reviewer comments and feedback.

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
