## [Reviewer comments · Microbiology Spectrum]

Microbiology Spectrum

Viral and cellular telomerase RNAs possess host-specific antiapoptotic functions

Ahmed Kheimar, Laetitia TRAPP-FRAGNET, Andele Conradie, Luca Bertzbach, Yu You, Mohammad Sabsabi, and Benedikt Kaufer

Corresponding Author(s): Benedikt Kaufer, Freie Universitat Berlin

Review Timeline:

Submission Date:	May 5, 2023
Editorial Decision:	June 5, 2023
Revision Received:	August 4, 2023
Accepted:	August 7, 2023

Editor: Donna Neumann

Reviewer(s): The reviewers have opted to remain anonymous.

Transaction Report:

DOI: <https://doi.org/10.1128/spectrum.01887-23>

June 5, 2023

Prof. Benedikt B. Kaufer
Freie Universität Berlin
Institut für Virologie
Robert von Ostertag-Straße 7-13
Berlin 14163
Germany

Re: Spectrum01887-23 (Viral and cellular telomerase RNAs possess host-specific antiapoptotic functions)

Dear Prof. Benedikt B. Kaufer:

Link Not Available

Sincerely,

Donna Neumann

Journals Department
Reviewer comments:

Reviewer #1 (Comments for the Author):

Summary:

Both viral and cellular telomerase RNAs (TR) have been implicated in promoting tumorigenesis, but the mechanism by which they do so is unclear. In this manuscript Kheimar et al., investigate the role of viral and cellular TRs in tumorigenesis using a Marek's disease virus model in chickens. The authors demonstrate that viral (v)TR, chicken (c)TR, and human (h)TR all share anti-apoptotic function but that this function appears to be species specific. The authors further imply that this species specific anti-apoptotic function may explain the apparent inability of hTR to compensate for vTR in promoting tumorigenesis.

For the most part, the manuscript is well-written and the experiments are relatively straightforward. The data demonstrating species-specific anti-apoptotic function is compelling, however, it is unclear from the tumorigenesis studies whether there is truly a species-specific effect on tumorigenesis due to the lack of an appropriate cTR control virus and a different timeframe of experiment compared to previous studies.

Major comment:

The rationale for looking at the potential species-specific anti-apoptotic functions for the various TRs and the implication that anti-apoptotic function of the TRs may be responsible for promoting tumorigenesis appears to be predicated on the conclusion that cTR and hTR differ in the ability to complement for the loss of vTR in promoting tumorigenesis. However, as presented, the data used to support this conclusion are not compelling.

Fig 3 does show disease incidence and tumor incidence are decreased in viruses expressing hTR or no TR compared to vTR (wt). However, a control virus expressing cTR is not included in these experiments.

The authors have previously published similar experiments with a virus in which vTR is replaced with cTR (ref. 14) from which they conclude "the insertion of the cellular TR from the chicken (cTR) completely restored the tumorigenesis of the virus (14)." (Lines 201-202). However, the data actually shown in ref 14 seem to suggest there is a delay in both disease incidence and tumor incidence in the cTR expressing virus compared to vTR expressing controls with tumor incidence only being fully restored at 91 days post infection which is substantially beyond the timeframe of the experiments shown in the current manuscript. At the 70 day post infection timepoint in the previous experiments (the endpoint of the experiments shown in the current manuscript), there appears to be a similar decrease in tumor incidence in cTR expression viruses compared to vTR expressing viruses as the defect they are showing here between hTR and vTR. Thus, without a direct comparison between cTR and hTR expressing viruses, it is unclear if there is really a difference between cTR and hTR in the ability to promote tumorigenesis or if both result in a delay in tumor formation compared to vTR and the shortened timescale of the experiment in the current study prohibits the authors from seeing complementation by hTR.

As hTR, cTR, and vTR have clear host-specific differences in the ability to inhibit apoptosis (Fig. 5), clarifying whether hTR and cTR actually have differences in the ability to promote tumorigenesis seems critical to appropriately interpreting what role (if any) the anti-apoptotic function may be playing in tumorigenesis in this model.

Minor comments:

Does the vTR mutant that cannot be incorporated into the telomerase complex previously identified by the authors (vTR P6.1, ref 20) maintain anti-apoptotic function? This would seem to be a relatively straightforward test the author's hypothesis that this function is telomerase-independent. In addition, if anti-apoptotic function is not maintained by this mutant, that would seemingly argue against anti-apoptotic function being important for tumorigenesis since this mutant retains the ability to promote tumorigenesis.

Statistical information is provided in the figure legends; however, with the exception of Fig. 3C and 3D, the samples being compared are unclear.

For clarity, Fig 5 should indicate which samples were treated with STS and which were not.

For the anti-apoptotic activity assays, the authors employ two different chicken T cell lines but use a human epithelial-like cell (293T). While the data are compelling that there are host-specific differences, matched cell types seems like it would be a better comparison.

Staff Comments:

Preparing Revision Guidelines

- Point-by-point responses to the issues raised by the reviewers in a file named "Response to Reviewers," NOT IN YOUR COVER LETTER.

- Upload a compare copy of the manuscript (without figures) as a "Marked-Up Manuscript" file.
- Each figure must be uploaded as a separate file, and any multipanel figures must be assembled into one file.
- Manuscript: A .DOC version of the revised manuscript
- Figures: Editable, high-resolution, individual figure files are required at revision, TIFF or EPS files are preferred

Please return the manuscript within 60 days; if you cannot complete the modification within this time period, please contact me. If you do not wish to modify the manuscript and prefer to submit it to another journal, please notify me of your decision immediately so that the manuscript may be formally withdrawn from consideration by Microbiology Spectrum.

To:
Editorial Board
Microbiology Spectrum

July 29, 2023

Dear Dr. Neumann,

We are delighted to have the opportunity to revise our manuscript titled “Viral and cellular telomerase RNAs possess host-specific antiapoptotic functions”, with the reference number: **Spectrum01887-23**. We have taken careful consideration of the valuable comments and recommendations provided by the reviewer, and we would like to outline how we have addressed their constructive criticism in our revised manuscript. Attached herewith, you will find a detailed point-by-point rebuttal.

We believe that the modifications we have made greatly enhanced the quality of the manuscript, and we sincerely hope that it will receive a favorable review and be deemed suitable for publication in Microbiology Spectrum. We are looking forward to hearing from you.

With best regards,

Benedikt B. Käufer, PhD

Point-by-point response to the reviewer's criticisms.

Reviewer #1

Major comment:

1. *The rationale for looking at the potential species-specific anti-apoptotic functions for the various TRs and the implication that anti-apoptotic function of the TRs may be responsible for promoting tumorigenesis appears to be predicated on the conclusion that cTR and hTR differ in the ability to complement for the loss of vTR in promoting tumorigenesis. However, as presented, the data used to support this conclusion are not compelling.*

We appreciate the time and effort you dedicated to reviewing our manuscript. The rationale for assessing potential anti-apoptotic activities is also based on the conserved secondary structure of TRs. The observation that vTR and cTR possess antiapoptotic activity is completely novel and likely plays an important role in MDV-induced tumorigenesis. This discovery alone is - in my point of view - one of the biggest steps we have made in our understanding how vTR (and TRs in general) can contribute to tumorigenesis in recent years. With respect to the species-specific nature of the anti-apoptotic functions, we performed additional experiments to confirm the species specificity as outlined in minor comment 4 (see also Figure 3).

2. *Fig 3 does show disease incidence and tumor incidence are decreased in viruses expressing hTR or no TR compared to vTR (wt). However, a control virus expressing cTR is not included in these experiments.*

Thank you for raising this interesting aspect. cTR was initially not part of this manuscript and was only included to validate the differences observed in the anti-apoptotic functions between vTR and hTR (chicken vs. human). With respect to the animal experiments, we previously assessed the cTR expressing virus (cTR instead of vTR in the RB-1B strain) using wt RB-1B virus as a control. In the current experiment, we directly compared the hTR expressing virus to this exact same reference virus. In addition, we had the vTR deletion virus (v Δ vTR) as a second control in both experiments, allowing us to determine if cTR and/or hTR compensate for the loss of vTR in MDV-induced tumorigenesis. These two control viruses used in both studies (cTR and hTR) allowed us to draw the respective conclusions. We performed additional data analyses to further address this and the next comment of the reviewer (see Comment 3 & Figure 1).

3. *The authors have previously published similar experiments with a virus in which vTR is replaced with cTR (ref. 14) from which they conclude "the insertion of the cellular TR from the chicken (cTR) completely restored the tumorigenesis of the virus (14)." (Lines 201-202). However, the data actually shown in ref 14 seem to suggest there is a delay in both disease incidence and tumor incidence in the cTR expressing virus compared to vTR expressing controls with tumor incidence only being fully restored*

at 91 days post infection which is substantially beyond the timeframe of the experiments shown in the current manuscript. At the 70 day post infection timepoint in the previous experiments (the endpoint of the experiments shown in the current manuscript), there appears to be a similar decrease in tumor incidence in cTR expression viruses compared to vTR expressing viruses as the defect they are showing here between hTR and vTR. Thus, without a direct comparison between cTR and hTR expressing viruses, it is unclear if there is really a difference between cTR and hTR in the ability to promote tumorigenesis or if both result in a delay in tumor formation compared to vTR and the shortened timescale of the experiment in the current study prohibits the authors from seeing complementation by hTR. As hTR, cTR, and vTR have clear host-specific differences in the ability to inhibit apoptosis (Fig. 5), clarifying whether hTR and cTR actually have differences in the ability to promote tumorigenesis seems critical to appropriately interpreting what role (if any) the anti-apoptotic function may be playing in tumorigenesis in this model.

Thanks for this important observation. Due to the Covid-19 pandemic, we had to terminate the hTR experiments earlier (70 dpi) than planned (91 dpi). To determine if cTR and/or hTR can complement the loss of vTR (and therefore contribute to tumorigenesis), the identical vTR deletion virus (vΔvTR) was used as a reference in both experiments as described above. To address the differences in the timescale, we compared the two animal experiments (cTR and hTR) at 70 days post infection. This analysis revealed that cTR significantly increased tumor incidence compared to vΔvTR at 70dpi (Fig. 1A). This was not observed for the vhTR at 70 dpi (Fig. 1B), suggesting that hTR does not complement the loss of vTR in MDV-induced tumorigenesis as highlighted in this manuscript. We also addressed this aspect in the discussion of the revised manuscript (line 204-211).

Figure 1. Comparison of tumor incidence in the animals infected with vcTR or vhTR from two in vivo experiments until 70 dpi. A) Tumor incidences in the animals infected with vcTR compared to those infected with vΔvTR until 70 dpi using a Kaplan-Meier analysis. B) Tumor incidences in the animals infected with vhTR compared to those infected with vΔvTR until 70 dpi using a Kaplan-Meier analysis. Fischer's exact test revealed no significant differences in the tumor incidences between vΔvTR and vhTR until

70 days post infection. In contrast, there was a significant difference in the tumor incidences between vΔvTR virus and vcTR ($p=0.034$, Bonferroni correction $p<0.0171$)

Minor comments:

1. Does the vTR mutant that cannot be incorporated into the telomerase complex previously identified by the authors (vTR P6.1, ref 20) maintain anti-apoptotic function? This would seem to be a relatively straightforward test the author's hypothesis that this function is telomerase-independent. In addition, if anti-apoptotic function is not maintained by this mutant, that would seemingly argue against anti-apoptotic function being important for tumorigenesis since this mutant retains the ability to promote tumorigenesis.

Thanks for pointing this out. We performed the proposed experiments and could demonstrate that the vTR P6.1 mutant significantly inhibits apoptosis in chicken T cells (Figure 2). As the P6.1 mutant is not part of this manuscript and outside the scope of the current manuscript, we will use the data in a future manuscript on the mechanism of vTR-mediated apoptosis inhibition.

519 **Figure 2. vTR P6.1 exhibits antiapoptotic activity in chicken T-cells.** The activation of the caspase pathways in chicken T cells (CU91 and 855-19 T) upon stimulation with 1μM STS. The caspase 3/7, caspase 8 and caspase 9 activity was measured in different cell lines and is shown as luminescence units for **A-C**) chicken T cells CU91, **D-F**) chicken T cells 855-19. CU91, 855-19 T cells harboring vTR, vTRP6.1, or hTR sequences were treated with 1uM STS for 3h at 40°C. Luminescence was normalized to background levels. Error bars represent the standard error of the mean of three independent experiments performed in duplicates (** $p<0.01$ assessed by 1-way ANOVA).

2. *Statistical information is provided in the figure legends; however, with the exception of **Fig. 3C and 3D**, the samples being compared are unclear.*

We included the requested statistical information in all figure legends as suggested.

3. *For clarity, **Fig 5** should indicate which samples were treated with STS and which were not.*

Thanks. All samples (except the DMSO control) were treated with STS. This is now indicated in the figure legend.

4. *For the anti-apoptotic activity assays, the authors employ two different chicken T cell lines but use a human epithelial-like cell (293T). While the data are compelling that there are host-specific differences, matched cell types seems like it would be a better comparison.*

Thank you for this valuable suggestion. As suggested, we evaluated the ability of hTR and vTR to inhibit apoptosis in a human T cell line (SupT1) and in chicken stem cell-derived epithelial-like cell line (ESCDL) [1]. Our apoptosis assays revealed that hTR can protect human Sup-T1 T cells from apoptosis. However, vTR cannot (Figure 3 A-C). The inverse activities was observed in chicken ESCDL cells (Figure 3 D-F). These data confirm that the vTR or hTR possess host-specific antiapoptotic function.

Figure 3. Antiapoptotic activity of vTR and hTR in human Sup-T1 and chicken ESCDL cells. Apoptosis was induced in Sup-T1 or ESCDL cells using 1 μ m STS. Caspase 3/7, caspase 8 and caspase 9 activation is shown as luminescence units for **A-C**) Sup-T1 cells **D-E**) ESCDL containing the empty vector (EV), vTR or hTR expression plasmid. Luminescence was normalized to background levels. Error bars represent the standard error of the mean of three independent experiments performed in duplicates. Asterisks indicate significant differences in comparison to empty vector control (**p<0.01 assessed by 1-way ANOVA).

References.

1. Vautherot, J.F., et al., *ESCDL-1, a new cell line derived from chicken embryonic stem cells, supports efficient replication of Mardiviruses*. PLoS One, 2017. **12**(4): p. e0175259.

August 7, 2023

Prof. Benedikt B. Kaufer
Freie Universität Berlin
Institut für Virologie
Robert von Ostertag-Straße 7-13
Berlin 14163
Germany

Re: Spectrum01887-23R1 (Viral and cellular telomerase RNAs possess host-specific antiapoptotic functions)

Dear Prof. Benedikt B. Kaufer:

Your manuscript has been accepted, and I am forwarding it to the ASM Journals Department for publication. You will be notified when your proofs are ready to be viewed.

Sincerely,

Donna Neumann
Editor, Microbiology Spectrum
